# Modeling and Analysis of a Reconfigurable Rover for Improved Traversing over Soft Sloped Terrains

**DOI:** 10.3390/biomimetics8010131

**Published:** 2023-03-22

**Authors:** Shipeng Lyu, Wenyao Zhang, Chen Yao, Zheng Zhu, Zhenzhong Jia

**Affiliations:** 1Department of Mechanical and Energy Engineering, Southern University of Science and Technology (SUSTech), Shenzhen 518055, China; 2Shenzhen Key Laboratory of Biomimetic Robotics and Intelligent Systems, Shenzhen 518055, China; 3Guangdong Provincial Key Laboratory of Human-Augmentation and Rehabilitation Robotics in Universities, Shenzhen 518055, China

**Keywords:** robot mobility, slope traversing, reconfigurable robot, terramechanics

## Abstract

Adjusting the roll angle of a rover’s body is a commonly used strategy to improve its traversability over sloped terrains. However, its range of adjustment is often limited, due to constraints imposed by the rover design and geometry factors such as suspension, chassis, size, and suspension travel. In order to improve the rover’s traversability under these constraints, this paper proposes a reconfigurable rover design with a two-level (sliding and rolling) mechanism to adjust the body’s roll angle. Specifically, the rolling mechanism is a bionic structure, akin to the human ankle joint which can change the contact pose between the wheel and the terrain. This novel adjustment mechanism can modulate the wheel–terrain contact pose, center-of-mass projection over the supporting polygon, wheel load, and the rover driving mode. Combining the wheel–load model and terramechanics-based wheel–terrain interaction model, we develop an integrated model to describe the system dynamics, especially the relationship between rover pose and wheel slippage parameters. Using this model, we develop an associated attitude control strategy to calculate the desired rover pose using particle swarm algorithm while considering the slip rate and angle constraints. We then adjust the sliding and rolling servo angles accordingly for slope traversing operations. To evaluate the proposed design and control strategies, we conduct extensive simulation and experimental studies. The results indicate that our proposed rover design and associated control strategy can double the maximum slope angles that the rover can negotiate, resulting in significantly improved traversability over soft sloped terrains.

## 1. Introduction

The exploration of outer space is an ongoing effort, and humanity has achieved numerous important milestones in recent years. Both China and the United States successfully landed their robotic rovers on Mars in 2021, respectively. The European Space Agency (ESA) is also on track with its own Mars exploration missions. An important task of planetary exploration is to find mineral resources of the planet and search for elements (e.g., water) that can sustain life. Autonomous robotic rovers are critical tools for these exploration missions.

In planetary exploration missions, scientists are often more interested in investigating certain areas, such as volcanic craters, where it can be more convenient to study the planet’s internal structure and mineral composition.

To reach these regions, the robots often need to traverse through sloped areas consisting of bedrock, gravel, and sand. These areas pose a great challenge to rover mobility and mission success. When the rover moves along the slope, the load on the downside wheel is much greater than that on the upside wheel due to the load transfer effect [1]. Hence, the downside wheel is more likely to exceed the maximum contact or traction force that can be provided by the soft soil (this constraint is closely related to the shear stress limit of the soil), thereby causing excessive side slip or sinkage [2], and may eventually lead to mission failure.

As the slope traversing task represents one of the core problems for the rover’s exploration tasks, many scholars have invested their efforts toward improving the rover’s traversability by shifting the rover’s center of mass (COM). There are two basic types of these COM-shifting methods (more details in Section 2): one is to change the robot pose, and the other one is to use the shifting of the robot arm on the rover body. These methods are limited by the rover’s size and suspension. We provide a novel design to overcome these limitations. The new configurable rover can adjust its wheelbase (via mode switching) to change its COM distribution. Furthermore, the rover can also change the contact pose between the wheel and the terrain. This function can reduce the risk of soil failure and ensure that the wheel–terrain contact model is more accurately used during this contact process.

In this paper, we focus on the slope traversing task shown in Figure 1. To solve this problem, a commonly used strategy is to use reconfigurable rovers (such as Ref. [2]) that can actively adjust its center of mass (COM) and attitude for improved wheel load balancing and slope traversability. Related studies are given in Section 2. Learning from the design of existing reconfigurable rovers, three contributions are provided in this paper as below:An actively reconfigurable rover design with a two-level adjustment mechanism is provided, and one of the mechanisms is a bionic structure that can adjust the contact angle. In addition, an integrated model based on pose and slippage parameters is introduced.An attitude control strategy for slope traversing based on particle swarm optimization (PSO) algorithm is provided. Based on this strategy, the rover can successfully traverse slopes under constraints ($\beta \leq 3^{\circ }$ & s≤0.2).Based on force and torque performance analysis during the slope traversing experiment, two optimization directions are provided for different slope angles, with each direction bearing its own unique benefit.

The method to solve the slope traversing task is described in Figure 1 and contains three steps. For the first step, we use on-board sensors such as IMU, camera (optical flow), and servo encoders to estimate the rover’s 6-DOF (degree-of-freedom) pose and velocity information; we also use the on-board camera for terrain perception, identifying terrain composition and geometry (e.g., slope angle). For the second step, we propose an attitude control strategy and use the integrated model consisting of the wheel–terrain force model and rover pose-wheel load model, to identify the desired rover pose (e.g., slip ratio *s*, slip angle β, and sinkage *h*) for optimal performance (e.g., minimizing rover slippage). We then calculate suitable motion mode and control parameters using inverse kinematics. During the traversing process, we keep the slip parameters of the rover within the bounds (s<0.2 & $\beta <3^{\circ }$). For the third step, we execute the motion control strategy.

## 2. Related Work

### 2.1. Reconfigurable Rover Design

Planetary rovers are expected to probe over steep sandy slopes, such as crater rims, where excessive wheel slippage and sinkage are critical issues. Ref. [3] provides a solution: mount linear stages on the rover’s suspension arms so that they can actively adjust the rover’s body pose (mainly the roll angle) for better adaption to the sloped terrain. The experiments show that both longitudinal and lateral slippages are greatly reduced by tilting the rover towards the uphill direction.

Similar to Ref. [3], the prototype of the reconfigurable rover Scarab in Ref. [4] could both change its height through a one-level adjusting mechanism, and also change its wheelbase.

Compared to the aforementioned studies, Ref. [5] proposes a novel rover design, which aims to use the rover’s rotating body to adjust the rover’s attitude through the roll and pitch rotation operations. The experiments show that the design can effectively improve the rover’s traversability on solid sloped surfaces. The soft terrain studies were evaluated in the study [6]. Furthermore, we herein present the modeling work associated with the rover design (similar to Refs. [7,8]), the details of which we have omitted since they are straightforward.

### 2.2. Environment Perception and State Estimation

During the exploration process, the main challenges for the rover are the unique terrain and granular medium. Thus, some researchers pursue obtaining that information online. For environment perception, such as slope detection, cameras or lidar are introduced into this field. Ref. [9] extended the concept of photometric stereo to photogrammetric-photometric stereo by incorporating collinearity equations into the imaging irradiance model, and higher-resolution and higher-precision surface reconstruction could be achieved. Some studies pay more interest to terrain physical properties estimation. There are two measures that are used mostly for online estimation: The model-based method (see Ref. [2]) and machine learning-based method (see Ref. [10]). Based on that online information of terrain, this research of rover state estimation and motion control has made tremendous progress which ensures rover’s safety motion (such as Ref. [11]).

### 2.3. Slope Traversing Strategy and Analysis

To solve the slope traversing problem, Ref. [12] proposes two control approaches for exploration rovers traversing sandy slope terrains. One method is a model-based feed-forward control using a characteristic diagram, called a thrust-cornering characteristic diagram. The other method is a sensor-based feedback control. One key to this feedback control is to compensate for three types of slippage, namely, the vehicle sideslip, the longitudinal and lateral slips of the wheel. Ref. [13] discusses the effects of attitude changes on downhill sideslip based on the slope failure mechanism and experiments on reconfiguring the rover attitude and wheel angles. The experiments show that the wheel–slope contact angle has a dominant effect on the sideslip when compared with that of readjusting the rover’s center of gravity. Refs. [14,15] proposes a robust path planning algorithm that can find a safe and efficient path based on a chance-constrained planning approach. This algorithm probabilistically guarantees safety again not immobilization, with a safety level specified by user-defined parameters. Learning from these studies, we introduced a novel pose control strategy for our reconfigurable rover.

## 3. Rover Modeling on Sandy Slope

We propose an integrated model consisting of a wheel–load model and a wheel–soil contact model for rover’s slope traversing. The wheel load model estimates the rover’s COM and the wheel load distributions, based on the rover’s attitude information. We adopt the wheel–soil contact model based on our previous work (Refs. [2,16,17]) to calculate the wheel forces generated by the terrain under different slip conditions. To simplify the problem, we make the following assumptions:The rover speed is very low; hence, the slope traversing can be considered as a quasi-static problem.The rover has four independently driven rigid wheels.The slope surface is flat and is uniformly covered with loose soil (we use loose sand in this paper).The slippage of each wheel during movement is the same, and is equivalent to the slippage of the rover.

### 3.1. Introduction of Adjustment Mechanism

In this part, we focus on showing the two-level adjustment mechanism in detail. As shown in Figure 2, the mechanism includes two components, the sliding-part and rolling-part, respectively.

Sliding-part: A coarse adjustment linkage that adjusts the wheel loads by changing the longitudinal distance between each wheel and the rover’s COM.Rolling-part: A fine adjustment linkage that can fine-tune the angle between the wheel’s side surface and slope surface or switch the rover’s driving mode. This mechanism is learned from the human ankle joint. As humans will change their foot pose while climbing the slopes, and this method is really helpful, we think our robot can also use this strategy to improve traversing ability.

### 3.2. Coordinate System Definition

To better describe the slope traversing problem, we establish three coordinate systems shown in Figure 2: the world coordinate system $\sum_{W}$, the vehicle body coordinate system $\sum_{B}$, and the wheel coordinate system $\sum_{C}$.

As shown in Figure 3, the world $z^{W}$ axis points in the upward direction (opposite to the gravity direction), while the $y^{W}$ axis points to the inner side of the slope. The $x^{W}$ axis is defined by the right-hand rule. We assume that the rover is heading towards the $x^{W}$ axis during slope traversing. The rover coordinate system $\sum_{B}$ can then be obtained through a rotation of $\sum_{W}$ about the $x^{W}$ axis by $\left(\alpha -\theta_{b}\right)$ angle in the uphill direction shown in Figure 2 and Figure 3, where α is the slope angle. The $x^{C}$ axis of the wheel coordinate system $\sum_{C}$ points to the travel direction of the wheels. The $y^{C}$ direction is perpendicular to $x^{C}$ and points to the upward slope direction. The $z^{C}$ axis is defined by the right-hand rule. The body coordinate system $\sum_{B}$ and the wheel coordinate system $\sum_{C}$ are fixed at the geometric centers of the rover and the wheel, respectively.

For easier analysis, we can divide the rover into three major components when estimating the 3D position of rover COM: main body, the sliding-part, and the rolling-part. Recall that the relative posture of driving wheel and the rolling-part does not change during adjustment; hence, we can add the driving mechanism to the group of rolling-part (see Figure 2).

Using the definition of physical parameters shown in Table 1, when adjusting the rover’s attitude, the main contributions come from $P_{S_{ij}}$ and $R_{R_{ij}}^{S_{ij}}$, whose parameterized representations are:(1)$$P_{S_{ij}}={\left[X_{S_{ij}},Y_{S_{ij}},Z_{S_{ij}}+0.21\varphi_{S_{ij}}\right]}^{T}$$
(2)$$R_{Rij}^{Sij}=\left[\begin{array}{ccc} 1 & 0 & 0\\ 0 & \cos\varphi_{R_{ij}} & -\sin\varphi_{R_{ij}}\\ 0 & \sin\varphi_{R_{ij}} & \cos\varphi_{R_{ij}} \end{array}\right]$$

Here, $\varphi_{S_{ij}}$ and $\varphi_{R_{ij}}$ represent the clockwise rotation angle of the sliding-part’s and rolling-part’s driving motor, respectively. According to the definition of COM, its expression in the world frame $\sum_{W}$ can be estimated by Equation (Equation 3), and $P_{com}^{W}$ is the COM representation in $\sum_{W}$.
(3)$$\left\{\begin{array}{c} P_{1}=T_{B}^{B}P_{B}m_{B}\\ P_{2}=\sum_{i=\{F,R\}}^{j=\{l,r\}}T_{S_{ij}}^{B}P_{S_{ij}}m_{S}\\ P_{3}=\sum_{i=\{F,R\}}^{j=\{l,r\}}T_{S_{ij}}^{B}T_{R_{ij}}^{S_{ij}}P_{R_{ij}}m_{R}\\ P_{com}^{W}=T_{B}^{W}\frac{P_{1}+P_{2}+P_{3}}{4\left(m_{S_{ij}}+m_{R_{ij}}\right)+m_{B}} \end{array}\right.$$

### 3.3. Wheel Load Model of the Reconfigurable Rover

As shown in Figure 3, $L_{d}$ represents the horizontal distance (in the world frame $\sum_{W}$) from the rover’s COM to the rolling-part’s rotation axis towards the downhill side of the slope, and $L_{u}$ represents the similar distance towards the uphill side of the slope. $H_{d}$ represents the vertical distance (in the world frame $\sum_{W}$) from the rover’s COM to the rotation axis of the rolling-part in the downhill side, and $H_{u}$ represents the vertical distance of the uphill side. *K* is the distance from the rotation center of the rolling-part to the geometric center of its connected wheels. $\theta_{b}$ is the angle between the $z^{B}$ axis of the rover’s body frame $\sum_{B}$ and the normal direction of the slope surface, $\theta_{d}$ is the rotation angle of the downhill side rolling-part to the $Y^{B}$ axis of the rover’s body frame $\sum_{B}$, while $\theta_{u}$ represents the angle in the uphill side. The loads on the uphill and downhill wheels can be expressed as:(4)$$W_{u}=\frac{(L_{d}\cos\alpha_{1}+K_{d1}-H_{d}\sin\alpha_{1}-K_{d2})W}{2\left(L_{d}+L_{u}\right)\cos\alpha_{1}-\left(H_{d}-H_{u}\right)\sin\alpha_{1}+K_{sum})}$$
(5)$$W_{d}=\frac{(L_{u}\cos\alpha_{1}+K_{u1}+H_{u}\sin\alpha_{1}+K_{u2})W}{2\left(L_{d}+L_{u}\right)\cos\alpha_{1}-\left(H_{d}-H_{u}\right)\sin\alpha_{1}+K_{sum})}$$
where:(6)$$\left\{\begin{array}{c} \alpha_{1}=\alpha -\theta_{b}\\ K_{d1}=K\cos\theta_{d}\cos\alpha_{1}\\ K_{d2}=K\sin\theta_{d}\sin\alpha_{1}\\ K_{u1}=K\cos\theta_{u}\cos\alpha_{1}\\ K_{u2}=K\sin\theta_{u}\sin\alpha_{1}\\ K_{sum}=K_{d1}+K_{u1}-K_{d2}+K_{u2} \end{array}\right.$$

### 3.4. Wheel–Soil Contact Model

Here we use slip rate *s* (assume s>0) and slip angle β to measure the rover’s slippage when moving on the slope:(7)$$\left\{\begin{array}{c} s=1-\frac{v_{x}}{r\omega }\\ \beta =\tan^{-1}\left(\frac{v_{y}}{v_{x}}\right) \end{array}\right.$$
where the slip rate *s* is the longitudinal slippage, $v_{x}$ represents the actual speed along the $x^{C}$ direction, *r* and ω represent the rover wheel radius and angular velocity, respectively. The slip angle β measures the slippage in the lateral direction. $v_{x}$ and $v_{y}$ represent the $x^{C}$ and $y^{C}$ velocities, respectively, as shown in Figure 2. Since the $x^{C}$ axis direction is the same as $x^{B}$, we take the slippage of wheel as the equivalent slippage of rover based on the assumptions above.

When the rover is moving on a loose slope, the wheel-terrain contact force can be divided into two parts: wheel load or supporting force $F_{b}$ provided by the bottom surface, and lateral force $F_{s}$ provided by the side surface. The total contact force on each wheel can be calculated as:(8)$$\left\{\begin{array}{c} F_{x}=F_{bx}+F_{sx}\\ F_{y}=F_{by}+F_{sy}\\ F_{z}=F_{bz}+F_{sz} \end{array}\right.$$

The wheel supporting force $F_{b}$ can be calculated based on Refs. [17,18,19,20], as follows:(9)$$\left\{\begin{array}{c} F_{bx}=r\int_{-\frac{b}{2}}^{\frac{b}{2}}\int_{\theta_{r}\left(y\right)}^{\theta_{r}\left(y\right)}\left\{\tau_{t}\cos\theta -\sigma \sin\theta \right\}d\theta dy\\ F_{by}=r\int_{-\frac{b}{2}}^{\frac{b}{2}}\int_{\theta_{r}\left(y\right)}^{\theta_{r}\left(y\right)}\tau_{l}d\theta dy\\ F_{bz}=r\int_{-\frac{b}{2}}^{\frac{b}{2}}\int_{\theta_{r}\left(y\right)}^{\theta_{r}\left(y\right)}\left\{\tau_{t}\sin\theta +\sigma \cos\theta \right\}d\theta dy \end{array}\right.$$
where, σ, $\tau_{l}$, and $\tau_{t}$ are functions of slip rate *s*, θ, and sinkage. Interested readers can refer to [17,18,19,20] for more details.

The contact force on the wheel sidewalls can be described in frame $\sum_{C}$ as following:(10)$$\left\{\begin{array}{c} F_{sx}=-\left(F_{sp}-F_{sa}\right)\sin\delta \sin\left|\beta \right|\\ F_{sy}=-\left(F_{sp}-F_{sa}\right)\cos\delta \sin\beta \\ F_{sz}=-\left(F_{sp}-F_{sa}\right)\sin\delta \sin\beta \end{array}\right.$$
where $F_{sp}$ and $F_{sa}$ are functions of slip angle β and sinkage. For more details, please refer to [2,21,22].

### 3.5. Integrated Model

As we have already completed the pose–load model and the wheel–terrain contact model, the relationship between rover pose and rover performance (slip ratio and slip angle) can be calculated or predicted. In Equation (Equation 11), the left is the 3-axis force acting on each wheel based on wheel–terrain contact model, and the right is the wheel load in 3-axis on each wheel based on pose-load model.
(11)$$\left\{\begin{array}{c} F_{x_{\left(i,j\right)}}=0\\ F_{y_{\left(i,j\right)}}=W_{\left(i,j\right)}\sin\alpha_{1}\\ F_{z_{\left(i,j\right)}}=W_{\left(i,j\right)}\cos\alpha_{1} \end{array}\right.$$

Based on the prior information of granular medium, geometric shape (slope angle in this paper) measured by on-board camera (see Figure 1), and rover pose measured by on board IMU, we could use the integrated model to estimate the slip parameters (slip ratio *s*, and slip angle β) which are the most important outcome parameters during the slope traversing process. As the equation of slip parameters is implicit, it is difficult to calculate slippage directly. Some optimization method are used to solve this problem.

## 4. Pose Control Strategy

The main objective of our pose control strategy is to ensure the safety of slope traversing with different inclinations, in other words, achieving better wheel load balancing, as discussed in Section 1. To this end, we propose a pose control strategy (Algorithm 1) to coordinate the outputs of multiple motors based on the slope angle and other parameters (robot and terrain data).

One important task of this algorithm is to select the operation mode (Figure 1) according to the slope angle. We define a threshold $max(\theta_{b})$ to be the slope angle that the rover can maintain wheel load balancing by adjusting the sliding-part alone. When the slope angle is less than $max(\theta_{b})$ ($16^{\circ }$ for our rover), the rover should use Mode I for slope traversing. Otherwise, it should use Mode II. After adopting suitable driving mode, the rover should take optimization pose to keep the slip parameters in constraints for slope traversing task. The second task is to select the wheel–slope contact pose (represented by the angle between wheel side surface and the slope surface) by adjusting the rolling-part according to the maximum contact force that the terrain can provide along the upside direction of the slope.
**Algorithm 1** Attitude control strategy for slope traversing. 1:**Input:** threshold $max(\theta_{b})$, and the max wheel-soil contact force $max(F_{y})$ along the $y^{\left(B\right)}$ axis. 2:**Output:** servo rotation angle $\varphi_{S_{ij}}$ and $\varphi_{R_{ij}}$ 3:**Initialization:** 4:acquire the slope angle α 5:**if** $\alpha ⩽max(\theta_{b})$ **then** 6:   choose operation state “Mode I” 7:**else** 8:   choose operation state “Mode II” 9:**end if**10:calculate the $\varphi_{S_{ij}}$ based on Algorithm 211:adjust the servos of the sliding-part $\varphi_{S_{ij}}$12:**if** $W_{\left(i,j\right)}sin\alpha ⩽max\left(F_{y}\right)$ **then**13:   to optimize wheel-slope contact pose14:**else**15:   to optimize wheel-slope contact force16:**end if**17:adjust the servos of the rolling-part $\varphi_{R_{ij}}$18:**return** current $\varphi_{S_{ij}}$ and $\varphi_{R_{ij}}$

**Algorithm 2** $\varphi_{S_{ij}}$ calculated by particle swarm optimization algorithm.
 1:
**Important Condition:**
 2:Particle position: $X=\left\{x_{i}|x_{i}=\varphi_{S_{ij}}\right\}$ 3:End condition: $\beta \leq 3^{\circ }$ & s≤0.2 4:Fitness:$f=\frac{\pi }{2\left(\varphi_{S_{ij_{now}}}-\varphi_{S_{ij_{last}}}\right)}+\frac{\pi }{2\beta }+\frac{1}{s}$ 5:
**Beginning:**
 6:initial starting parameters 7:calculate the fitness 8:find out the optimization 9:update position and velocity of particle10:**if** not meet the end condition **then**11:   go to step 712:
**else**
13:   save $\varphi_{S_{ij}}$14:
**end if**
14:**return** $\varphi_{S_{ij}}$


In Algorithm 1, lines 5–11 correspond to the coarse adjustment process by the sliding-part, while line 12–17 correspond to the fine adjustment process by the rolling-part. As shown in Algorithm 1, when the rover encounters a slope, it can detect the slope angle α using on-board sensors. First, we choose the motion mode by comparing the actual slope angle α and the threshold $max(\theta_{b})$. Then, we reconfigure the rover through coarse and fine adjustments by changing the sliding-part and rolling-part, respectively. We use the particle swarm optimization (PSO) in Algorithm 2 to find out the $\varphi_{S_{ij}}$. During fine adjustment, we need to make a simple judgment on the terrain properties of the slope, that is the downward component force $W_{\left(i,j\right)}\sin\alpha$ and the maximum upward contact force $F_{y}$ provided by the slope. If the former is larger, we use the rolling-part to optimize wheel-soil contact pose. Otherwise, we use the rolling-part to optimize wheel-soil contact force.

There are two benefits using this control strategy. First, we can choose the most suitable driving mode for slope traversing tasks. Then, sub-optimal pose of rover could be calculated based on PSO algorithm. Second, we can adjust the wheel’s roll angle by tuning the rolling-part, thereby meeting different control or optimization objectives (see Section 5.2.2 for details).

## 5. Simulations and Experiments

In this section, we verify the proposed rover design through slope traversing experiments on loose soil. As shown in Figure 2, our rover ROMA-Sloper is driven by 12 servo motors; the gear-and-rack sliding mechanism has a travel distance about 10 cm. In experiments, we use Realsense cameras (on-board T265/D435i fixed in $\sum_{B}$, a D435i fixed in world frame $\sum_{W}$) to measure the 3D trajectory and velocity of the rover. We also record the force data of each wheel using 6-axis F/T sensor (vendor: SRI, model: M3813D), as shown in Figure 2. Table 2 lists the specifications of the rover testbed. We use the MATLAB 2022b to do simulation and the terrain (soil) information are learned from Ref. [2]. As shown in Figure 1, the soft terrain slope is created by a sandbox with a size of 1.2 m (length) × 1.0 m (width) × 0.15 m (depth). The slope of the sandbox can be changed by adjusting the linear actuator under the sandbox. The adjusting range of the slope is $0^{\circ }$ to $32^{\circ }$. To keep the independence of each experiment, we keep the sand plane in the box flat manually.

### 5.1. Slope Traversability Analysis

In this section, we give a simulation result to show the efficiency of the COM adjusting ability of this design in Figure 2. We analyse the display relation of the rover’s COW, COB, and COM while driving along the slope. In common sense, when COM and COB overlap, the load of each wheel of the rover is equal. As shown in Figure 2, we calculate the position of COB, COM, and COW in $Y^{W}$ axis. It is worth noting that some references (e.g., [21]) use COB instead of the actual COM to estimate the wheel load. This approximation works fine for bilaterally symmetric (the left and right sides are symmetric) rovers, e.g., ROMA-Sloper in Mode I and the rover in [21]. As shown in Figure 4a,b, the differences between the $COM_{y}$-$COW_{y}$ intersection and the $COB_{y}$-$COW_{y}$ intersection are quite small. However, if the rover is not bilateral symmetric, for example, ROMA-Sloper in Mode II shown in Figure 1, Figure 2 and Figure 3, this approximate method will generate large error. This can be seen from Figure 4c,d, where the $COM_{y}$-$COW_{y}$ intersection and the $COB_{y}$-$COW_{y}$ intersection is significantly different.

To verify the efficacy of the introduced algorithm, we study experimental and simulated rover slippages shown in Figure 5. As shown in Figure 5a,b, the rover achieves the best performance on a $10^{\circ }$ slope when it is operated in Mode I and its roll angle equals $10^{\circ }$; the results agree well with the estimation in Figure 4a. As shown in Figure 5c,d, the rover achieves the best performance on a $30^{\circ }$ slope when it is operated in Mode II and its roll angle equals $8^{\circ }$; the results agree well with the estimation in Figure 4d. These results validate the desired COM position estimation. There are multiple factors that cause the sim-to-real differences shown in Figure 5. The main reason is that the simplified ideal wheel-soil interaction model we used in the simulation cannot quite capture the influences of multiple varying factors such as the complex 3D contact geometry, the granular medium fluidization. These phenomena could also be seen in other works such as Ref. [2].

Figure 6 plots the rover’s actual configuration and roll angle under different slope angles, based on multiple successful experiments (5 iterations for every slope angle). It should be noted that this value (maximum slope angle traversed in Mode I) is actually limited by the travel distance of the rover’s sliding mechanism. In Ref. [2], the rover can traverse a $20^{\circ }$ slope with a very long-travel sliding mechanism. Figure 6 also indicates that switching to Mode II can greatly improve the rover’s traversability. The rover can traverse a $30^{\circ }$ slope, which almost doubles the maximum angle ($max(\theta_{b})$) when driving in Mode I. The maximum slope angle for Mode II is limited by the distance between the rolling joint and the slope surface. If this distance is too small, the rolling joint will penetrate the soil surface (see the top right figure in Figure 6), thereby affecting the traversing significantly. Hence, we can traverse even deeper slopes if using larger wheels.

### 5.2. Force Characteristics

In the sequel, we analyze the wheel’s force characteristics during coarse and fine adjustments.

#### 5.2.1. Coarse Adjustment

In this section, we show through experiment the importance of mode selection and its effect on wheel forces and traversing trajectory. To this end, we select $20^{\circ }$ slope because it is larger than the threshold ($16^{\circ }$) for mode switching. The experiment results are shown in Figure 7. We see that by switching the operation mode, the rover can improve its slope traversability significantly.

As shown in Figure 4 (initial state I), the rover’s COM projection onto the $Y^{W}$ axis is closer to the downhill wheel compared against the uphill wheel when operated under Mode I. Consequently, the wheel load and driving torque of the downhill wheels are larger than those of the uphill wheels. Hence, we need to adjust the sliding-part to even the wheel load distributions. However, this adjustment is limited by the travel range of the sliding-part. As shown in Figure 7, for Mode I, the *z*-axis force difference between the uphill and downhill wheels is around 8N. There is also some difference in the driving torque. The rover has an obvious downhill slip, its slip angle is $10.5^{\circ }$. To solve this problem, we need to switch to Mode II. The force difference is further reduced to 1.8 N, i.e., 77.5% reduction compared to Mode I. The side slip of the rover is negligible.

#### 5.2.2. Fine Adjustment

As shown in Algorithm 1, there are two control objectives (force versus pose/angle) during the fine adjustment process by tweaking the rolling-part. In the sequel, we analyze the force and interaction characteristics, and the applicable scenarios of these two objectives. As shown in Figure 8, our rover can traverse the $10^{\circ }$ slope efficiently when using different optimization objectives. Although they have very similar trajectories, their force characteristics such as the supporting force ($F_{z}$), lateral force ($F_{y}$), and driving torque are different. For the pose optimization, the downside force of the y-axis is greater than the force optimization. Therefore, its downside force will more easily overcome the limitation of the supporting force of the soil in the y-axis. Thus, we need to choose a suitable optimization direction for different situations. For small slope angles, we prefer the angle optimization, i.e., we adjust the wheel pose so that the wheel side surface is normal to the slope surface. As shown in see Figure 6, we select angle optimization when the slope angle is smaller than $12^{\circ }$ (determined by the physical characteristics of the slope medium). The reason is that the angle optimization helps to mitigate the discrepancy problems in Ref. [21], when one trying to apply terramechanics models or parameters developed for flat ground to the sloped case. Another benefit is that we can have more even wheel load distributions ($F_{z}$).

Note that the magnitude of the maximum lateral force is larger under pose/angle optimization when compared with the force optimization, although it is still within the shear threshold of the terrain. However, if the slope angle keeps increasing, the lateral force will exceed this threshold, thereby causing significant downhill slip (i.e., large slip angle). In this case, we should switch to force optimization, in order to reduce the lateral force. Consequently, the wheel side surface will be parallel to gravitational force, as shown in Figure 8b. This operation can help to reduce the “slope failure” risk [2].

We implemented several experiments of the rover for slope traversing tasks. When the rover traverses the slope successfully, we note its pose (roll angle of the rover body) and its motion mode. We drew the relation between the slope angle and rover pose in Figure 6. As shown in Figure 6, compared to the $12^{\circ }$ slope case, the wheel side surface is almost parallel with the gravity direction during $16^{\circ }$ slope traversing. For even steeper slopes, the rover switches to Mode II and uses force optimization during fine adjustment process. As shown in Figure 6, the wheel side angle is close (but not exactly) to the vertical direction. This is because during the rolling process, the rolling joint will penetrate into the soil surface if the wheel side surface is perfectly aligned with the gravity direction. This can be mitigated if we use larger wheels. A better approach is to design a remote rolling adjustment mechanism, i.e., place the rolling joint far away from the slope surface, to avoid physical interference.

## 6. Conclusions

In order to improve the slope traversability over soft terrains, this paper proposes a novel rover design with a two-level (sliding and rolling) adjustment mechanism. We derive the traversing model by integrating the wheel–load model and wheel–soil interaction model. We then develop the associated control strategy to modulate the rover pose to maximize its traversing performance. We also evaluate the efficacy through simulations and experiments. For small slope angles, we can adjust the rover’s pose so that the rover’s wheel axes are parallel to the slope surface. This can help mitigate the sim-to-real discrepancy problem discussed in the literature when trying to apply terramechanics models that are developed for flat surfaces to use on sloped surfaces. For large slope angles, the control strategy (mode II) will rotate all wheels to the downhill side. This helps to shift the rover’s COM to the uphill direction (relative to the rover supporting polygon), thereby achieving better wheel balancing for improved slope traversing. For our future work, we plan to augment the current rover design with an independent wheel-steering mechanism [12], so that we can combine two control authorities provided by the two-level adjustment mechanism (focusing on changing COM) and the steering mechanism (focusing on vehicle dynamics). To examine to the fullest extent, we will investigate a wheel–leg hybrid (as presented in [23]), where we can fully exploit the contact mechanics for the goal of reaching ultimate slope mobility. Some control policies based on reinforcement learning for slope traversing tasks will be implemented so that the rover can handle more challenging traversing situations.

## Figures and Tables

**Figure 1 biomimetics-08-00131-f001:**
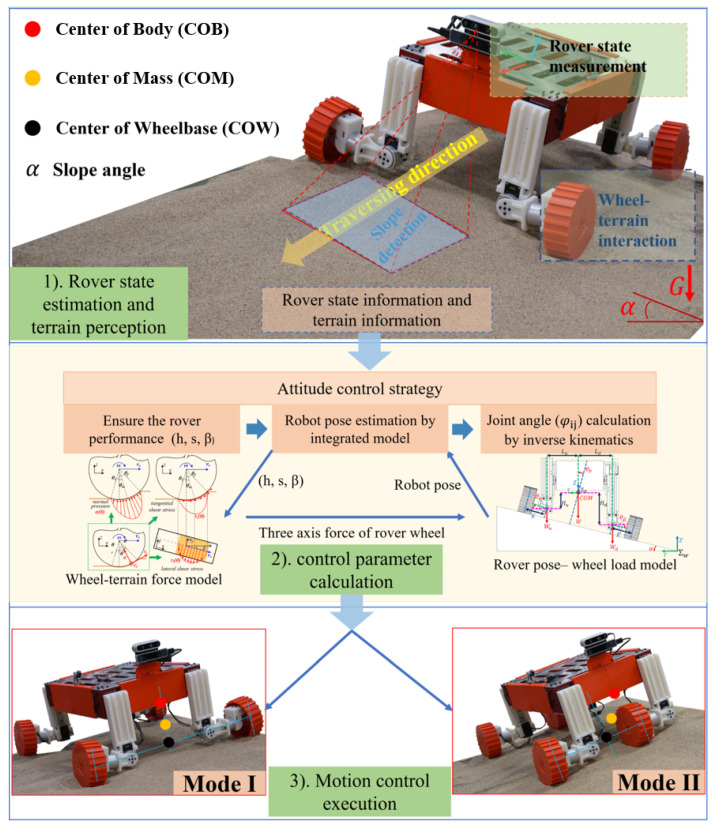
Introduction to the soft terrain slope traversing task using our ROMA-Sloper rover as an example. This task can be divided into three parts: (1) rover state estimation and terrain perception, (2) rover attitude control parameter calculation, and (3) motion control execution. ROMA-Sloper has 4 wheels, 12 motors, 4 F/T sensors and an Intel T265 camera (with built-in visual odometry function) and an Intel D435i RGB-D camera for terrain perception.

**Figure 2 biomimetics-08-00131-f002:**
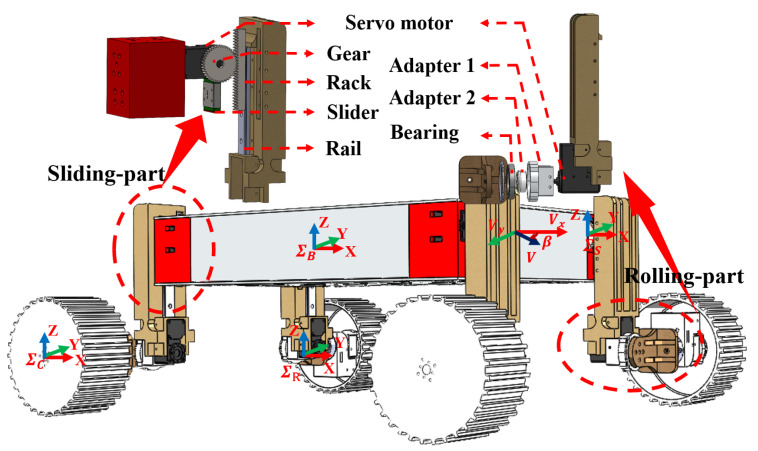
The ROMA-Sloper rover is reconfigurable, and it contains an active two-level adjustment mechanism: sliding-part and rolling-part. The detailed structures of the two parts are shown in the exploded view. Some coordinate systems are attached to this rover.

**Figure 3 biomimetics-08-00131-f003:**
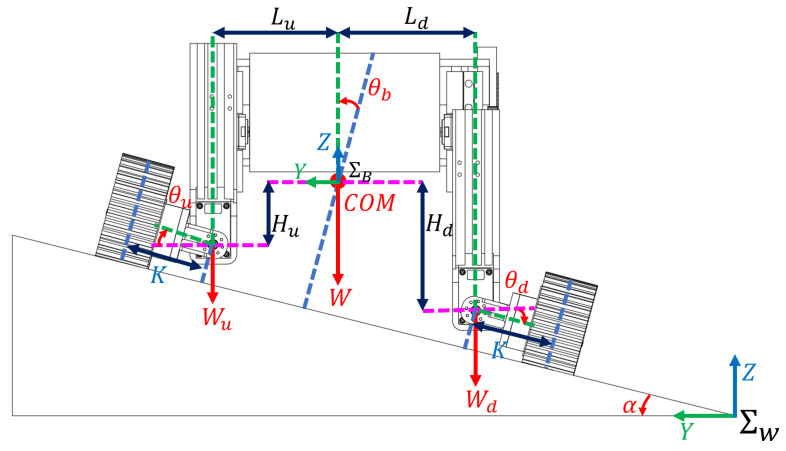
Geometrical parameter definitions for slope traversing. Some important distance notices are marked in the simplified rover figure. The COM’s position and wheel load can be calculated by these notices.

**Figure 4 biomimetics-08-00131-f004:**
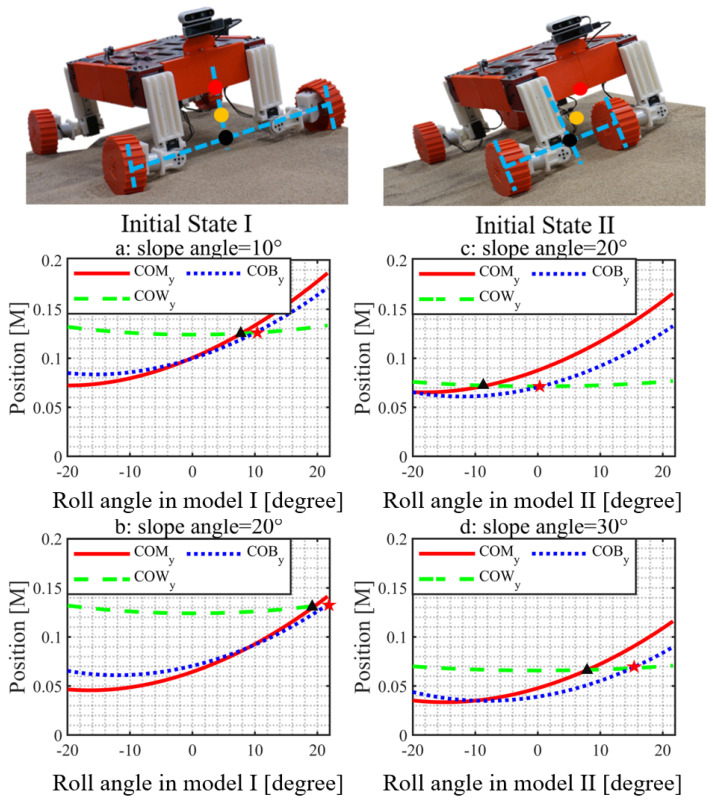
The relations between the rover roll angle and the COM, COB, and COW positions in $Y^{W}$ direction when rover operates in different driving modes. The black triangle refers to the cross of green dash line ($COW_{y}$) and red solid line $\left(COM_{y}\right)$; The red star refers to the cross of green dash line ($COW_{y}$) and blue dotted line ($COB_{y}$). For situation a, the condition is that the rover traverses $10^{\circ }$ slope in mode I, while the slope angle is $20^{\circ }$ for situation b. For situation c, the condition is that the rover traverses $20^{\circ }$ slope in mode II, while the slope angle is $30^{\circ }$ for situation b.

**Figure 5 biomimetics-08-00131-f005:**
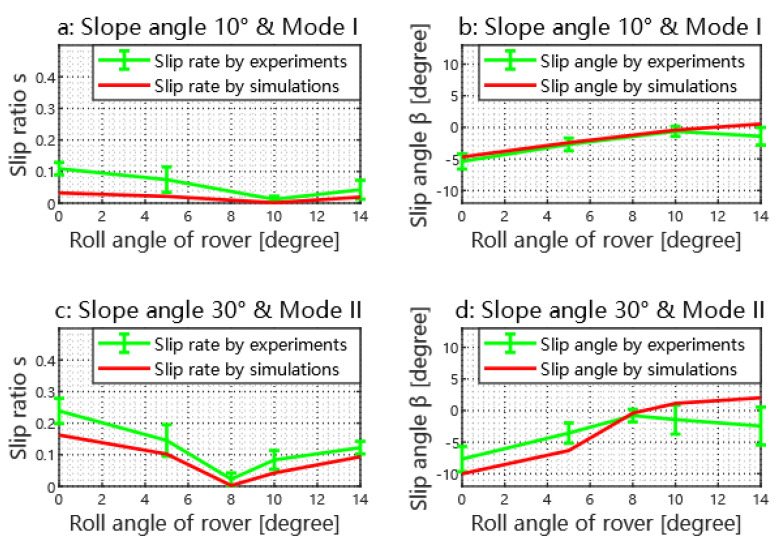
Experimental and simulation results of rover slippages during $10^{\circ }$ and $30^{\circ }$ slope traverses. The wheel’s angular velocity is 20 rpm. Sub-figures (**a**,**b**) show the slip ratio and angle for the rover traversing on a $10^{\circ }$ slope in mode I, while the conditions of sub-figures (**c**,**d**) are the mode II and $30^{\circ }$ slope. We measure the slip parameters of the rover as it traverses the slope at different flow angles.

**Figure 6 biomimetics-08-00131-f006:**
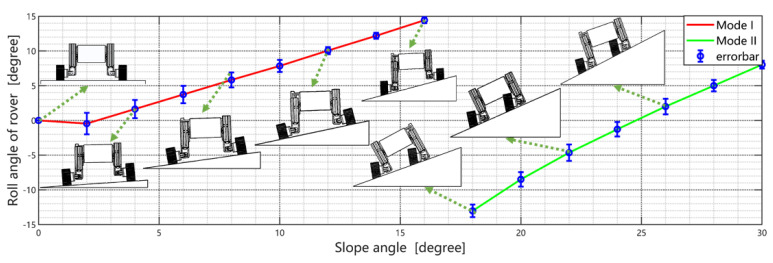
The rover’s optimal configuration and roll angle under different slope angles. The left part (red line) corresponds to Mode I, and the right part (green line) corresponds to Mode II. The slope angle threshold $max(\theta_{b})$ between Mode I and Mode II is about $16^{\circ }$.

**Figure 7 biomimetics-08-00131-f007:**
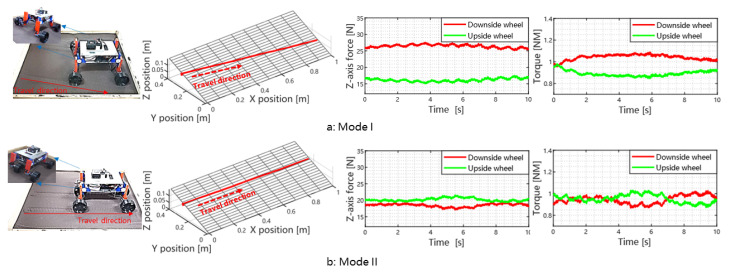
The rover trajectory and force/torque data were collected in the traversing task on a $20^{\circ }$ slope. For sub-figure (**a**), the rover state is the mode I, While mode II is selected for the rover in sub-figure (**b**). The first column shows the experimental scenarios and the motion modes of ROMA-Sloper. The second column shows the trajectory of ROMA-Sloper after traversing the slope. The last two columns show the force and torque information of the wheel during the traversing process.

**Figure 8 biomimetics-08-00131-f008:**
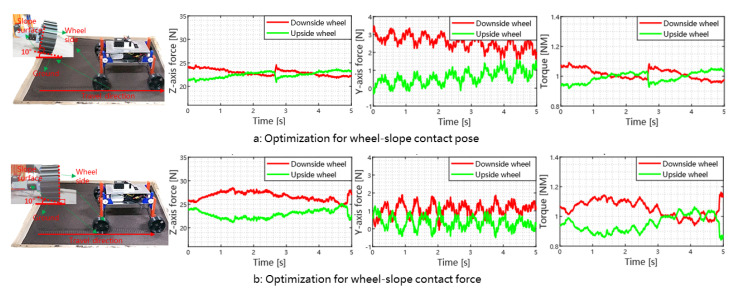
The rover trajectory and force/torque data (expressed in wheel frame $\sum_{C}$) for different optimization objectives in the fine adjustment process. The first column shows the experimental scenarios, and the wheel-terrain contact states are shown as sub-figures (**left**-**top**) in detail. The last three columns show the force and torque information of the wheel during the traversing process. In sub-figure (**a**), the rover’s wheels are parallel to the slope (contact pose optimization); In the sub-figure (**b**), the rover’s wheels are parallel to the direction of gravity (contact force optimization).

**Table 1 biomimetics-08-00131-t001:** Symbols used in the rover model.

Symbol	Definition
$\sum_{B}$	Rover’s body coordinate system
$\sum_{S_{ij}}$	Sliding-part (i,j) coordinate system
$\sum_{R_{ij}}$	Rolling-part (i,j) coordinate system
$R_{S_{ij}}^{B}$	Rotation matrix, rotation relation of $\sum_{S_{ij}}$ relative to $\sum_{B}$
$R_{R_{ij}}^{S_{ij}}$	Rotation matrix, rotation relation of $\sum_{R_{ij}}$ relative to $\sum_{S_{ij}}$
$d_{S_{ij}}^{B}$	Origin coordinate, origin relation of $\sum_{S_{ij}}$ relative to $\sum_{B}$
$d_{R_{ij}}^{S_{ij}}$	Origin coordinate, origin relation of $\sum_{R_{ij}}$ relative to $\sum_{S_{ij}}$
$P_{B}$	COM’s positions of the rover’s body in $\sum_{B}$
$P_{S_{ij}}$	COM’s positions of the sliding-part in $\sum_{S_{ij}}$
$P_{R_{ij}}$	COM’s positions of the rolling-part in $\sum_{R_{ij}}$
$m_{B}$	Mass of rover’s body (3.1 kg)
$m_{S}$	Mass of single sliding-part (0.6 kg)
$m_{R}$	Mass of single rolling-part (1.0 kg)

(i,j) is wheel numbering, i∈{F,R},j∈{L,R}. E.g., {F,R} represents the front right wheel.

**Table 2 biomimetics-08-00131-t002:** Specifications of the rover testbed.

Parameters	Value	Parameters	Value
Size (mm)	L600 × W540 × H230	Mass (kg)	9.5
Wheel size (mm)	ϕ140 × W50	Tread (mm)	490
Wheel base (mm)	460	Gravitational acceleration (m/s${}^{2}$)	9.81

## Data Availability

No new data were created or analyzed in this study. Data sharing is not applicable to this article.

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
