# Peer review of "Modeling and Analysis of a Reconfigurable Rover for Improved Traversing over Soft Sloped Terrains"

_biomimetics, 2023, doi:10.3390/biomimetics8010131_

Round 1
Reviewer 1 Report
Content
-L107/108 is that assumption valid? couldn't for example the rover's front wheels have more slippage than back wheels? --> an arc is introduced into the trajectory
- from the design it looks like you have really low ground clearance by introducing the ankle mechanism. Would it make sense to integrate that DoF at shoulder level?
- by switching to mode II it seems that you introduce a higher roll to the body (fig8). Would that not counteract your goal? In mode I it looks like trying to keep the roll angle of the central body low, which seems to be intuitive. By switching to Mode II, you need space below the rover for the wheel and hence to roll it more. I think this is not nessecarily a good solution
- my impression is, that Mode II indeed shifts the COM towards the uphill boundary of the support polygon. However, this is, because the support polygon is getting smaller. Yet, I agree that obviously the uphill wheels will take more of the load share by this maneuver.
- the last sentence of conclusion with aiming for wheel-leg hybrid: this is a good idea. You should then have a more close look on state of the art, at least more close than presented in this paper (see also comments on bibliography provided below)
- The whole system is quite small and lightweight. When aiming for a "real space rover" you should state how a higher mass might influence the seemingly fragile ankle mechanism. Certainly, there will be problems in scaling up the rover size (current Mars rovers have a mass of several hundrets of kg)
- it would be good to have a mass distribution: how much mass is allocated in the body, how much in the legs/wheels? If a considerable amount of mass is close to the wheels, it might be a good idea to even push the uphill wheels further up the hill. Especially when considering a later implementation with active legs and a bigger system more approaching a "real space rover"
Typos/Grammar etc
L21: human have --> humanity has?
L61: the robot is called Scarab, not Scara
Figures
Fig1:
- mixed acronyms and description of CoM and CoB
- Center of Mess --> Center of Mass
- 2x red dot as symbol CoW should presumably be black
- parameters (h,s,\beta) are not described/explained (they are introduced in ll155-160
Fig4:
- you should use the same scaling on the y axes to have the chats better comparable
Fig7: it says "for different optimization objectives", yet subcaptions a and b are identical. also, better style would be to split up the image and use LaTeX's subfigure environment...
Bibliography
There are duplicate entries 2./3. and 17./18.
There might be several relevant systems with active suspension missing:
- ATHLETE and Tri-ATHLETE rovers, e.g.
Wheeler et al. (2010) FootSpring: A Compliance Model for the ATHLETE Family of Robots, i-SAIRAS'10
- Mammoth rover, e.g.
Reid et al (2017) Complex Manoeuvres for the Wheel-on-Leg Planetary Analogue MAMMOTH Rover, 68th International Astronautical Congress
- SherpaTT rover, e.g.
Cordes et al (2018) Design and field testing of a rover with an actively articulated suspension system in a Mars analog terrain, Journal of Field Robotics
- RoboSimian, e.g.
Reid et al (2020) Autonomous Navigation over Europa Analogue Terrain for an Actively Articulated Wheel-on-limb Rover (IROS'20)
Reviewer 2 Report
Authors develop an integrated model to describe the system dynamics, especially the relationship between rover pose and wheel slippage parameters. there are few comments
the caption of Figure 1 is as pragraph. it is suggested to write short caption and move the explaination to text in the body of paper.
Most recent work should be addressed such as Survey on artificial intelligence based techniques for emerging robotic communication, Convergence of machine learning and robotics communication in collaborative assembly: mobility, connectivity and future perspectives and etc.
figure 3 and figure 8 needs more explanation
explain with details the software used for simulation
Reviewer 3 Report
I am really grateful to review this manuscript. In my opinion, this manuscript can be published once some revision is done successfully. In order to improve the rover’s traversability under various constraints, this study proposes a reconfigurable rover design with a two-level (sliding and rolling) mechanism to adjust the body’s roll angle. Notably, this study combines the wheel-load model and wheel-soil interaction model to describe the system dynamics, especially the relationship between rover pose and wheel slippage parameters. I would argue that this is a rare achievement. However, it can be noted that reinforcement learning can aid in the further improvement of the wheel-steering mechanism in this study. Reinforcement learning is a branch of machine learning in which (1) the environment presents a series of rewards, (2) an agent (e.g., rover) takes a series of actions to maximize the cumulative reward in response, and (3) the environment moves to the next period with given transition probabilities. I would like to suggest the authors to address this issue (possible application of reinforcement learning) in detail in the section of Conclusion.
Round 2
Reviewer 2 Report
this paper proposes a novel rover design with a two-level (sliding and rolling) adjustment mechanism. the authors derive the traversing model by integrating the wheel-load model and wheel-soil interaction model. it is good work
Authors should highlight the motivation and how to prove the novel of work
Authors should update the related work such as Survey on artificial intelligence based techniques for emerging robotic communication, Convergence of machine learning and robotics communication in collaborative assembly: mobility, connectivity and future perspectives, and Autonomous Multi-Robot Collaboration in Virtual Environments to Perform Tasks in Industry 4.0
verify eq.3, 9
explain the environements of experiments with more details
reorder the figures in right way fig 4, 5, 6, 7,
add more explaination with details in figure 4, 7
